# Preventing the Increase in Lysophosphatidic Acids: A New Therapeutic Target in Pulmonary Hypertension?

**DOI:** 10.3390/metabo11110784

**Published:** 2021-11-17

**Authors:** Thomas Duflot, Ly Tu, Matthieu Leuillier, Hind Messaoudi, Déborah Groussard, Guillaume Feugray, Saïda Azhar, Raphaël Thuillet, Fabrice Bauer, Marc Humbert, Vincent Richard, Christophe Guignabert, Jérémy Bellien

**Affiliations:** 1UNIROUEN, INSERM U1096, CHU Rouen, Department of Pharmacology, Normandie University, F-76000 Rouen, France; vincent.richard@univ-rouen.fr (V.R.); jeremy.bellien@chu-rouen.fr (J.B.); 2INSERM UMR_S 999, Hôpital Marie Lannelongue, F-92350 Le Plessis-Robinson, France; lyieng@gmail.com (L.T.); raphael.thuillet@inserm.fr (R.T.); marc.humbert@aphp.fr (M.H.); christophe.guignabert@inserm.fr (C.G.); 3School of Medicine, Université Paris-Saclay, Le Kremlin-Bicêtre, F-92290 Châtenay-Malabry, France; 4UNIROUEN, INSERM U1096, Normandie University, F-76000 Rouen, France; leuillier.matthieu@gmail.com (M.L.); hindiimessaoudi@gmail.com (H.M.); deborah.groussard@laposte.net (D.G.); saida.azhar1@univ-rouen.fr (S.A.); 5UNIROUEN, INSERM U1096, CHU Rouen, Department of General Biochemistry, Normandie University, F-76000 Rouen, France; guillaume.feugray@chu-rouen.fr; 6UNIROUEN, INSERM U1096, CHU Rouen, Department of Cardiology, Normandie University, F-76000 Rouen, France; fabrice.bauer@chu-rouen.fr

**Keywords:** lysophospholipids, lysophosphatidic acids, cardiovascular diseases, HPLC-MS/MS, rodent models, pulmonary hypertension, chronic heart failure, hypertension

## Abstract

Cardiovascular diseases (CVD) are the leading cause of premature death and disability in humans that are closely related to lipid metabolism and signaling. This study aimed to assess whether circulating lysophospholipids (LPL), lysophosphatidic acids (LPA) and monoacylglycerols (MAG) may be considered as potential therapeutic targets in CVD. For this objective, plasma levels of 22 compounds (13 LPL, 6 LPA and 3 MAG) were monitored by liquid chromatography coupled with tandem mass spectrometry (HPLC/MS^2^) in different rat models of CVD, i.e., angiotensin-II-induced hypertension (HTN), ischemic chronic heart failure (CHF) and sugen/hypoxia(SuHx)-induced pulmonary hypertension (PH). On one hand, there were modest changes on the monitored compounds in HTN (LPA 16:0, 18:1 and 20:4, LPC 16:1) and CHF (LPA 16:0, LPC 18:1 and LPE 16:0 and 18:0) models compared to control rats but these changes were no longer significant after multiple testing corrections. On the other hand, PH was associated with important changes in plasma LPA with a significant increase in LPA 16:0, 18:1, 18:2, 20:4 and 22:6 species. A deleterious impact of LPA was confirmed on cultured human pulmonary smooth muscle cells (PA-SMCs) with an increase in their proliferation. Finally, plasma level of LPA(16:0) was positively associated with the increase in pulmonary artery systolic pressure in patients with cardiac dysfunction. This study demonstrates that circulating LPA may contribute to the pathophysiology of PH. Additional experiments are needed to assess whether the modulation of LPA signaling in PH may be of interest.

## 1. Introduction

Cardiovascular diseases (CVD) are the leading cause of mortality and a major contributor to disability. In 2019, 523 million cases and 18.6 million deaths were reported corresponding to a 100% and a 50% increase since 1990, respectively. Since CVD remain the leading cause of disease burden in the world and still rise in all countries, the need to find new therapeutic target is of utmost importance [1]. CVD strongly rely to atherosclerosis and both are characterized by an increase in oxidative stress and chronic low-grade inflammatory condition associated to lipid homeostasis dysregulation [2]. The effectiveness of therapeutic strategies based on circulating lipids reduction such as statins or PCSK-9 inhibitors confirms the close link between atherosclerosis, CVD and lipid metabolism and it is now well admitted that those strategies significantly reduce long-term mortality and morbidity, especially in the elderly [3,4,5]. Most lipids circulate through bloodstream, either free or bound to other molecules including albumin or lipoproteins. Lipoproteins are macromolecular complexes of lipids and proteins with a central hydrophobic core containing non-polar lipids such as triglycerides and cholesterol esters surrounded by a hydrophilic layer composed of phospholipids, free cholesterol and apolipoproteins. CVD trigger the production of oxidized low-density lipoprotein (ox-LDL) and oxidized lipoprotein(a) (oxLp(a)) which become highly enriched in oxidized phospholipids (ox-PL) displaying pro-atherogenic properties [6,7,8]. Ox-PL are especially prone to hydrolysis by the lipoprotein associated phospholipase A_2_ (Lp-PLA_2_), also known as serum platelet-activating factor-acetyl hydrolase (PAF-AH), an inflammatory marker of CVD, leading to the release of both oxidized fatty acids and lysophospholipids (LPL) [9,10]. These compounds exert deleterious biological effects due to their interaction with nucleic acids, phospholipids and proteins promoting atherosclerosis and cardiovascular events [11]. Interestingly, bioactive LPL, and in particular lysophosphatidic acids (LPA) have potent effects on vascular cells, promoting vacoconstrictor, pro-inflammatory and proliferating effects that are known to contribute to atherosclerosis [12,13]. However, the role of LPL in the pathophysiology of many other CVD has been poorly evaluated [14,15,16,17]. LPA metabolism is complex with various anabolic and catabolic pathways. There are four major enzymatic pathways for LPA production: the extracellular LPL-autotaxin (LPL-ATX) pathway (1), the phosphatidic acid-phospholipase pathway (PA-PLA_1_/PLA_2_) (2), the monoacylglycerol kinase (MAGK) pathway (3) and the *de novo* glycerophosphate acyltransferase (GPAT) pathway (4). Amongst these pathways responsible for LPA production, the last three are intracellular and reversible with the intervention of lysophosphatidic acid acyltransferases (LPAAT), lipid phosphate phosphatases (LPP) and lysophospholipases (lysoPL) for pathway (2), (3) and (4), respectively (Figure 1) [18].

Thus, this study aims to evaluate circulating LPL, LPA and MAG levels in different rat models of CVD, i.e., hypertension (HTN), heart failure (HF) and pulmonary hypertension (PH) in order to provide new insights to the interest of targeting LPA metabolism with pharmacological compounds.

## 2. Results

### 2.1. Angiotensin-II Induced Hypertension (Ang-II HTN)

As expected, administration of Ang-II for 28 days induced an increase in aldosterone level (242 ± 108 vs. 1030 ± 633 pM, *p* = 0.004; Figure 2A) and in systolic blood pressure (SBP) (117 ± 4 vs. 176 ± 11 mm Hg, *p* < 0.001; Figure 2B) compared to control rats. Ang-II HTN also led to a decrease in left ventricular fractional shortening (LVFS) (45 ± 3 vs. 26 ± 3%, *p* < 0.001; Figure 2C) and an increase in heart weight (1.54 ± 0.13 vs. 1.76 ± 0.13 g, *p* = 0.004, Figure 2D), showing the development of cardiac dysfunction and hypertrophy.

Amongst the 22 monitored compounds, four LPA species exhibited significant increase in the HTN group compared with control group (LPA(16:0), LPA(16:1), LPA(18:1) and LPA(20:4)). However, these increases were no longer statistically significant after Benjamini–Hochberg (BH) correction to control the false discovery rate (Table 1) [19].

### 2.2. Chronic Heart Failure (CHF)

Definitive coronary artery ligation-induced CHF was defined by the decrease in LVFS (54 ± 4 vs. 26 ± 5% for control and CHF groups, respectively, *p* < 0.001, Figure 3A), the increase in heart weight (1.48 ± 0.12 vs. 1.68 ± 0.15 g, *p* = 0.008, Figure 3B), which concerned both the left (1.13 ± 0.09 vs. 1.27 ± 0.13 g, *p* = 0.016, Figure 3C) and right (0.24 ± 0.04 vs. 0.30 ± 0.05 g, *p* = 0.009, Figure 3D) ventricles but no change in mean pulmonary artery pressure (17.1 ± 1.1 vs. 17.0 ± 1.4 mm Hg, *p* = 0.861).

Regarding LPA metabolism, LPA(16:0), LPE(16:0) and LPE(18:0) were increase in CHF rats compared to control rats, whereas LPC(18:1) was decrease. However, these differences were no longer significant after BH correction (Table 2).

### 2.3. Pulmonary Hypertension (PH) Induced by Sugen/Hypoxia (SuHx)

Administration of the VEGF-receptor antagonist sugen (SU5416) associated with chronic hypoxia was used to induce severe PH. The SuHx rat model is a well-recognized animal model of severe PH (Group 1 PH). In this model, mean arterial pulmonary pressure (mPAP) was markedly increased compared with the control group (48 ± 7 vs. 16 ± 1 mm Hg, *p* < 0.001, Figure 4A). Heart weight was also increased (1.7 ± 0.2 vs. 1.2 ± 0.1 g, *p* < 0.001, Figure 4B), which not concerned the left (0.99 ± 0.06 vs. 0.96 ± 0.10 g, *p* = 0.472, Figure 4C) but only the right (0.73 ± 0.14 vs. 0.25 ± 0.03 g, *p* < 0.001, Figure 4D) ventricle. In addition, cardiac output was decreased in the PH group compared with the control group (53 ± 9 vs. 98 ± 4 mL/min, *p* < 0.001, Figure 4E).

PH was associated with an increase in all monitored LPA that remains significant after BH correction except for LPA (18:0), and a trend tower an increase in LPE(16:0) with an adjusted *p*-value of 0.056 (Table 3 and Figure 5).

### 2.4. Impact of LPA on Pulmonary Artery Smooth Muscle Cell (PA-SMC) Proliferation

To better evaluate whether LPA species may be involved in the pathophysiology of PH, we investigated their impact on the proliferation of human pulmonary artery smooth muscle cells (PA-SMC) derived from PH and control patients using 5-bromo-2-deoxyuridine (BrdU) incorporation (Figure 6A,B, respectively). Of interest, the conducted ANOVA was significant (*p* = 0.006 and *p* < 0.001, respectively) and all tested LPA species (LPA(18:1), LPA(18:2) and LPA(20:4)) were found to increase PA-SMC proliferation, a phenomenon that was more pronounced in PH PA-SMC.

### 2.5. Relationship between LPA Levels and Index of PH in Patients with Cardiac Dysfunction

We aimed to determine if LPA levels are associated with estimated pulmonary artery systolic pressure (PAPs) determined by transthoracic echocardiography in 90 patients with cardiac dysfunction followed in the Department of Cardiology of Rouen University Hospital. The estimated PAPs was 36 ± 14 mm Hg and its increase was positively associated with LPA(16:0) levels (*p* = 0.013; Figure 7).

## 3. Discussion

### 3.1. LPL, LPA and MAG Quantitation

The present analytical method aimed to monitor 22 compounds. The choice has been made to investigate compounds with and without analytical standards to have the widest possible overview of LPA metabolism based on known fragmentation patterns for each class. This leads to the limit that results are expressed as compound-to-IS area under the curve ratio and not as absolute quantitation. Furthermore, since we did not have specific deuterated internal standard for each compound, we cannot be sure of the reliability of the correction of matrix effects. However, for each model, all samples were prepared and analyzed uninterruptedly resulting in the absence of inter-batch variability [20]. Furthermore, the use of these IS, even if not being the best-suited, allowed to decrease within-batch variability normalizing area under the curve of the monitored compounds. Moreover, the low dispersibility of the intra-group values allows us to be sufficiently confident about the robustness and reliability of the results obtained. Finally, coefficients of variation for all monitored compounds did not exceed 30% for both repeatability and reproducibility except for LPC(22:6), LPE(18:0), MAG(18:1) and MAG(20:4).

Precautions should be taken in the preparation and analysis of these compounds. First of all, since LPA are highly acidic compounds, it is strongly recommended to perform an acidic extraction to avoid poor recovery. This could be carried out using hydrochloric or formic acid. Here, we used formic acid to prevent production of LPA by LPL hydrolysis that could be induced by too acidic solutions. Furthermore, it is mandatory to chromatographically resolve LPA and LPL that possess the same esterified fatty acid since collision-induced dissociation may artificially produce LPA. The natural abundance of LPA being a tiny fraction of LPL, artificial production or detection of only a minor part of LPL to LPA, either during extraction or MS analysis, could dramatically increase the concentration of these compounds [21].

In our study, LPL and MAG were monitored in plasma samples mainly reflecting extracellular production by ATX and distribution through lipoproteins and albumin. Since LPA are also constitutive of oxLDL, quantitation of albumin and lipoproteins, especially oxLDL, as well as expression and/or activity of key enzymes involved in LPA metabolism could have been worth to better assess relationships within LPA metabolism

### 3.2. CVD Models

All evaluated models have been well characterized with an increase in SBP, a decrease in LVFS and an increase in mPAP for the HTN, CHF and PH model, respectively. As observed in humans, the three tested models were also associated with the development of cardiac hypertrophy and dysfunction. Of note, although the development of PH secondary to left heart disease is frequent, we did not observe an increase in mPAP three months after the surgery used to induce CHF in rats.

The development of HTN induced a significant increase of 3 LPA species under unadjusted assumption. This may be justified by the fact that Ang-II modulates LPA1 receptor function [22] and induces LPA production by phospholipases activation as well as the production of phosphatidic acids [23,24]. Of note, Ang-II also enhances cell proliferation and hypertrophy [25].

CHF rats exhibited an increase in LPA(16:0) and LPE(16:0-, 18:0-) associated with a decrease in LPC(18:1). All increased compounds possess an esterified saturated fatty acid (SFA) suggesting a link between CHF-mediated oxidative stress and desaturase activity [26,27]. Furthermore, it is now well known that reducing intake of saturated fatty acids may reduce LDL-c and CVD risks [28,29]. SFA possessed numerous deleterious effects such as inflammation, apoptosis, mitochondrial dysfunction and oxidative stress [30] but their impact has mainly been evaluated as non-esterified fatty acids [31]. Thus, the role of esterified SFA in LPA and LPL needs further investigations. Finally, we used an ischemic model of CHF induced by coronary ligation since myocardial infarction is the most frequent cause of CHF in humans. However, it must be noted that CHF may developed following hypertension, diabetes, kidney disease, valve diseases or congenital heart disease.

In the SuHx rat model, severe PH was induced and LPA metabolism and activity seem to be a promising pathway to target. Indeed, 80% of the monitored LPA exhibited a 2- to 3-fold increase when compared with the control group. Furthermore, exogenous addition of LPA species in a cell media of cultured human PA-SMC at a concentration of 1 μM, which is relevant when compared with circulating LPA levels [32], led to PA-SMC proliferation. In fact, accumulation of PA-SMCs within the pulmonary arterial walls is one of the most prominent features of PH, which can lead to the narrowing or occlusion of pulmonary vessels and therefore plays an important role in the occurrence and development of PH [33,34,35,36]. At this time, the role of LPA remains unclear even if their increase in hypoxic pulmonary vascular remodeling has already been described [37]. Of note, LPA are able under physiological conditions to regulate the hypoxia-inducible factor 1α (HIF-1α) [38], that can interact with enzymes and other transcription factors in order to control vascularization and tissue growth in response to hypoxic conditions [39]. Furthermore, we observed a positive association between 16:0 LPA level, which is the most concentrated LPA specie in plasma, and the estimated PAPs in patients with cardiac dysfunction.

Although the results we obtained strengthen the hypothesis that LPA exert deleterious effects in PH, contradictory results have been reported in the literature with either protective [40,41] or harmful [42,43,44] effects of these lipid mediators against lung inflammation and fibrosis. Thus, blocking LPA synthesis by autotaxin or LPA actions using specific receptor antagonists in the model of SuHx represents a next step to finally demonstrate the deleterious role of LPA in PH development.

Of note, it may have been interesting to assess in parallel the evolution of LPL in a model of atherosclerosis since LPA species accumulate in high concentrations in atherosclerotic lesions contributing to thrombosis development [45,46].

### 3.3. LPA Signaling

Since LPA metabolism is complex with distinct anabolic and catabolic pathways, it may be easier to specifically act on LPA receptors rather than try to modulate enzymes involved in LPA metabolism regulation. However, it is now well admitted that LPA activate at least 6 specific G protein-coupled receptor named LPAR_1_–_6_. The downstream signals derived from activation of LPAR involve Rho, phospholipase C, phosphatidylinositol 3-kinase and adenylate cyclase intracellular pathways, thus producing diverse physio(patho)logical effects [18]. Recently, LPAR_4_ was found to contribute to elevation of blood pressure with the more potent effect for LPA 18:1 and 20:4 species [47]. Interestingly, LPAR activation exhibited different reactivity according to the acyl chain (carbon length and number of insaturations) and the esterification position (sn-1 or sn-2) of the LPA [48]. Another key point is related to the relative expression of those receptors in lung tissue where PH takes place. All LPAR_1_–_6_ were expressed in lung tissue according to https://www.genecards.org/ (accessed on 20 August 2021) and to date, it has only been demonstrated that LPAR_1_ and LPAR_2_ knock-down protects against pulmonary remodeling, lung inflammation and lung injury [40,49,50].

## 4. Materials and Methods

### 4.1. Experimentation

#### 4.1.1. Chemicals

Methanol (MeOH), Dichloromethan (DCM) and water of HPLC grade were purchased from Carlo Erba (Fontenayaux-Roses, France). Formic acid was purchased from VWR chemicals (Leuven, Belgium). LPA(16:0), LPA(17:1), LPC(17:1), LPE(17:1) LPA(18:1), LPC(16:0), LPC(18:1), LPE(16:0) LPS(16:0) and MAG(18:1) were purchased from Avanti Polar Lipids (Interchim, Montluçon, France). Ammonium acetate and formic acid were purchased from Carlo Erba (Fontenay-aux-Roses, France). Chromatographic Accucore XL C18 4 μm (150 × 3 mm) column was purchased from ThermoFisher Scientific (Illkirch-Graffenstaden, France)

#### 4.1.2. Animals

All the animal care and procedures were approved by French Animal Experimentation Ethics Committees and performed in accordance with the guidelines from the French National Research Council for the Care and Use of Laboratory Animals (Permit Numbers: Apafis #24107 and #11484). All experiments were performed in 10-weeks-old male wild-type Sprague-Dawley (SD/Crl) (Charles-River) rats.

### 4.2. LPL, LPA and MAG Quantitation

#### 4.2.1. Blood Sampling and Processing

At the time of sacrifice, blood samples were drawn in the aorta of rats anesthetized with isoflurane using 2-mL syringes. Blood was transferred on a prechilled ethylenediaminetetraacetic acid (EDTA) tubes and immediately centrifuged 5 min at 4500× *g* (4 °C). Then, the plasma was frozen in liquid nitrogen and stored at −80 °C until analysis. For LPA, LPL and MAG quantitation, 300 μL of plasma were thawed at room temperature, spiked with internal standards (IS) and subjected to protein precipitation using 1 mL of MeOH. Then, liquid-liquid extraction was performed after the addition of 100 μL of HCOOH (10%), 2 mL of DCM and 350 μL of H_2_O. The bottom organic phase was collected after being mixed and centrifuged (4500× *g*, 10 min, +4 °C), evaporated to dryness under a gentle stream of nitrogen and reconstituted in 100 μL of MeOH.

#### 4.2.2. High Performance Liquid Chromatography (HPLC)

HPLC was carried out using a Prominence Shimadzu UFLC system consisting of a DGU-20A3 degasser, a LC-20AB pump, a SIL-20ACHT autosampler and a CTO20AC oven (Shimadzu, Prominence, Kyoto, Japan). Chromatographic separation was performed on an Accucore XL C18 column (4 μm particle size, 150 mm length × 3 mm inner diameter). The autosampler temperature was set at +8 °C, the column oven at 50 °C, the injected volume was 10 μL, and the flow rate was 500 μL/min. The method had the following gradient conditions using 1% CH_3_COOH and 10 mM ammonium acetate in MeOH (solvent A) and 1% CH_3_COOH and 10 mM ammonium acetate in water (solvent B): 0–0.5 min, 70% A, 0.5–5.5 min, 70–95% A, 5.5–10 min, 95% A, 10–10.5 min, 95–70% A, 10.5–12 min, 70% A.

#### 4.2.3. Tandem Mass Spectrometry (MS^2^)

MS^2^ was performed using a 4500 QTRAP operating in the positive/negative electrospray ionization (ESI) switching mode (Sciex, Toronto, Canada). Instrument control and data acquisition were performed with Analyst v1.6.3 software. The source parameters were optimized as follow; ion spray voltage: −4500 V and +4500 V for negative and positive ionization mode, respectively; nebulisation gas: 60 psi; desolvatation gas: 50 psi; curtain gas: 30 psi; source temperature: 500 °C, entrance potential: −10 V and +10 V for negative and positive ionization mode, respectively, collision activation dissociation: medium. Relative quantitation was carried out using multiple reaction mode (MRM) to monitor transition from precursor to product ions. MS^2^ parameters were optimized by direct infusion of LPA(16:0), LPA(17:1), LPA(18:1), LPC(16:0), LPC (17:1), LPC(18:1), LPE(16:0), LPE(17:1) and MAG(18:1) in pure MeOH at a concentration of 1 μg/mL. For monitored compounds without analytical standards (LPA(18:0-; 18:2-; 20:4-; 22:6-); LPC(16:1-; 18:0-; 18:2-; 20:4-; 22:6-); LPE(18:0-; 18:1-; 18:2-; 20:4-; 22:6-) and MAG(18:2-; 20:4-), transitions used were obtained from the LIPID MAPS tools “Predict MS/MS spectrum for a glycero(phospho)lipid” (https://www.lipidmaps.org/tools/) (accessed on 18 August 2021). According to typical ionization and fragmentation patterns of each LPL class, MS1 were [M–H]^−^ adduct and MS2 was set to 152.9 for LPA (Glycerol-3-phosphate ion with loss of H_2_O), MS1 were [M-CH_3_]^−^ adduct and MS2 was set to the carboxylate ion of the corresponding fatty acid (sn1 acyl chain [RCOO]^−^) for LPC, MS1 were [M–H]^−^ adduct and MS2 was set to 196.0 (neutral loss of fatty acid) for LPE and MS1 were [M+H]^+^ adduct and MS2 was set to the acylium form of the corresponding fatty acid (sn1 acyl chain ([RC=O]^+^)) for MAG (Table 4). Chromatograms showing the elution of the monitored compounds, calibration curves of the analytical standards used and repeatability and reproducibility analysis using pooled plasma performed in triplicate on two separate days are given in Appendix A.

### 4.3. Murine Models of Cardiovascular Diseases (CVD)

#### 4.3.1. Angiotensin-II Induced Hypertension (HTN)

Systemic arterial hypertension was induced using 4-week Ang-II infusion with osmotic pumps (0.25 μg/kg/day), implanted subcutaneously in isoflurane-anesthetized rats. Non-implanted rats served as controls. Tail-cuff plethysmography was used in trained conscious animals to confirm the development of systemic hypertension.

#### 4.3.2. Chronic Heart Failure (CHF) Induced by Coronary Artery Ligation

For ischemic HF, myocardial infarction was induced by definitive left coronary artery ligation as previously described [51]. Briefly, rats were anesthetized using methohexithal (50 mg/kg i.p.), intubated and ventilated at 60 cycles/min (tidal volume 1 mL/100 g of body weight). A left thoracotomy was performed, and the heart exposed. A 6/0 polypropylene suture was passed around the proximal left coronary artery, which was tied in order to induce myocardial ischemia. In this case, 10 min after coronary artery ligation, the chest was closed, the pneumothorax was evacuated, and the animals were allowed to recover. Sham-operated rats, subjected to the same protocol except that the coronary artery was not occluded, served as controls. in this case, 12 weeks after surgery, the development of HF was confirmed using transthoracic echocardiography performed in rats anesthetized with isoflurane, using a Vivid 7 ultrasound echograph (GE Healthcare, Buc, France). A two-dimensional short axis view of the left ventricle was obtained at the level of the papillary muscle, in order to record M-mode tracings. Left ventricular end-diastolic (LVEDD) and systolic diameters (LVESD), allowing the determination of LV fractional shortening (FS) as FS (%) = ((LVEDD-LVESD)/LVEDD) × 100.

#### 4.3.3. Pulmonary Hypertension (PH) Induced by Sugen Hypoxia (SuHx)

For PH, the SuHx model, in which occlusive neointimal lesions are observed as in humans, was used as previously described [33]. Briefly, rats were injected subcutaneously with SU5416 (a VEGF-receptor antagonist; 20 mg/kg) and exposed to hypoxia (10% FiO_2_) for 3 weeks. Then, these rats returned to normoxia (21% FiO_2_) for additional 5 weeks before evaluation. Control rats were not injected and remained under normoxia for 8 weeks.

To confirm the development of PH, right ventricular hemodynamic measurements were performed in anesthetized rats using a polyvinyl catheter introduced into the right external jugular vein, advanced in the right ventricle, and further, in the pulmonary artery, allowing the measurement of mean pulmonary arterial pressure.

### 4.4. Human PA-SMC Proliferation

Lung specimens were obtained during lobectomy or pneumonectomy for localized lung cancer (control) and after lung transplant due do pulmonary hypertension (PH). For control group, lung specimens were collected at a distance from the tumor foci. This study was approved by the local ethics committee (ID CPP Est-III: 18.06.06; ID RCB: 2018-A01252-53) and all patients gave informed consent before the study. Human PA-SMC were isolated and cultured as previously described [33,34]. PA-SMC proliferation was measured by BrdU incorporation and by cell counting. BrdU staining was measured by the DELFIA^®^ Cell proliferation kit (PerkinElmer, Courtaboeuf, France) and Time-resolved fluorometer EnVisionTM Multilabel Reader (PerkinElmer) 24 h of culture without and with LPA(18:1), LPA(18:2) and LPA(20:4) at 1 μM.

### 4.5. Chronic Heart Failure Patients

In this case, 90 patients with cardiac dysfunction were included during a consultation visit in the Department of Cardiology of Rouen University Hospital. Blood sampling was performed allowing the quantitation of monitored compounds as described in Section 4.2. and estimated pulmonary artery systolic pressure, an indicator of PH, was obtained using transthoracic echocardiography. A value higher than 40 mm Hg is usually considered as abnormal and should prompt further investigation for PH diagnosis using right heart catheterization. The study was approved by the local ethics committee (ID RCB: 2014-A00834-43) and all patients gave written informed consent before the study.

### 4.6. Data and Statistical Analysis

Data, statistical analysis and captions were performed using R v4.1.0 software [52] and DescTools v0.99.43 package [53]. For relative quantitation of LPL, data were expressed as median (interquartile range or IQR) and p-values were computed by *t*-test and adjusted by Benjamini and Hochberg for multiple test corrections in order to decrease the false discovery rate [19]. Captions were performed using ggplot2 v3.3.5, ggsci v2.9 and ggpubr v0.4.0 packages [54,55,56]. For cell culture data analysis, ANOVA was conducted first then *p*-values were computed using Dunnett’s post-hoc test. For LPL, LPA and MAG quantitation, due to the lack of analytical standard for several compounds, 17:1- derivatives of each class (LPA, LPC and LPE) served as internal standard (IS) and data were expressed as compound-to-IS area under the curve (AUC) ratio that is unit-free. For MAG, 17:1-LPC was used as IS. R code for statistical analysis, rat models’ data, cell culture data and human raw data are available in Appendix A, respectively.

## 5. Conclusions

This study suggests that an increase in circulating LPA, observed in the relevant SuHx model and in patients with cardiac dysfunction, may contribute to the pathophysiology of PH. Additional experiments are warranted using modulators of LPA metabolism and signaling such as specific inhibitors, receptor antagonists or gene deletions in particular in the SuHx model to finally demonstrate that targeting LPA pathway may help to decrease occurrence and/or improve the outcome of PH.

## Figures and Tables

**Figure 1 metabolites-11-00784-f001:**
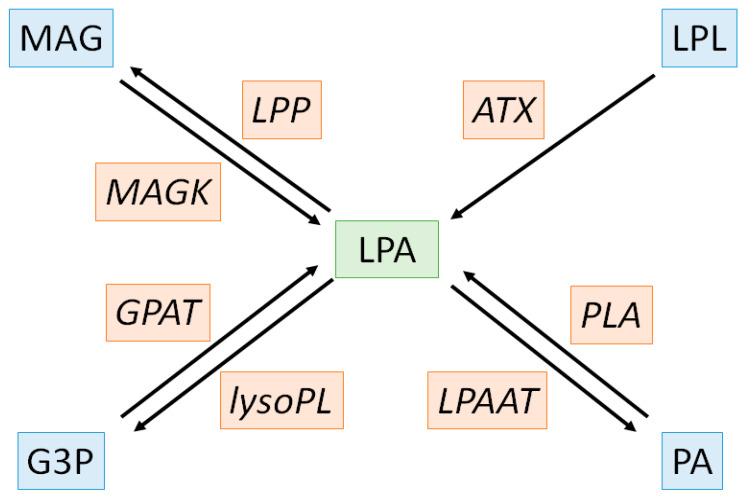
Biochemical pathways of lysophosphatidic acid metabolism. LPA, substrates/products and enzymes are depicted in green, blue and orange, respectively. ATX: autotaxin, G3P: glycero-3-phosphate, GPAT: glycerophosphate acyltransferase, LPA: lysophosphatidic acid LPAAT: lysophosphatidic acid acyltransferases, LPL: lysophospholipids, LPP: lysophosphatidic acid phosphatase, lysoPL: lysophospholipase, MAG: monoacylglycerol, MAGK: monoacylglycerol kinase, PA: phosphatidic acid, PLA: phospholipase.

**Figure 2 metabolites-11-00784-f002:**
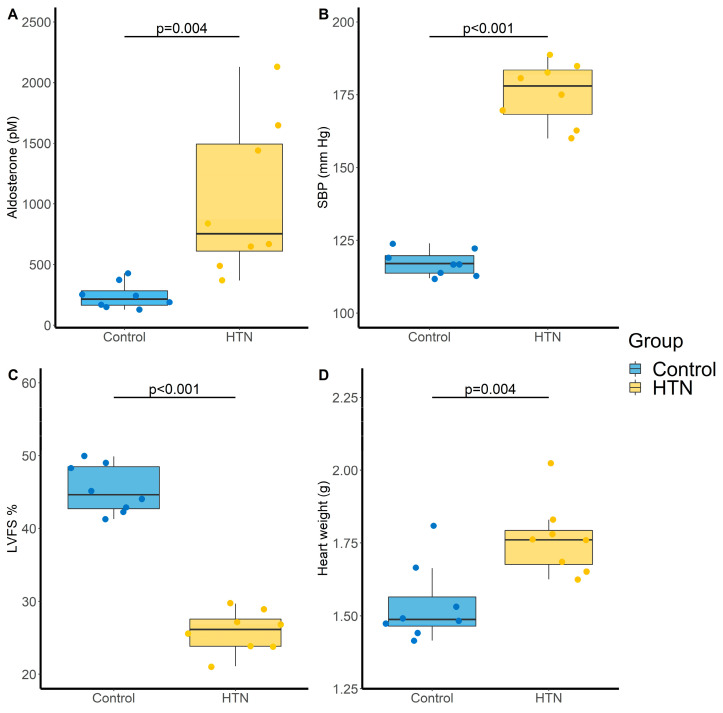
Evaluation of cardiovascular parameters in control and hypertensive (HTN) rats. (**A**) Aldosterone levels (pM); (**B**) Systolic blood pressure (SBP, mm Hg); (**C**) Left ventricular fractional shortening (LVFS, %) and (**D**) Heart weight (g). N = 8 for each group. *p*-values computed by Student’s *t*-test.

**Figure 3 metabolites-11-00784-f003:**
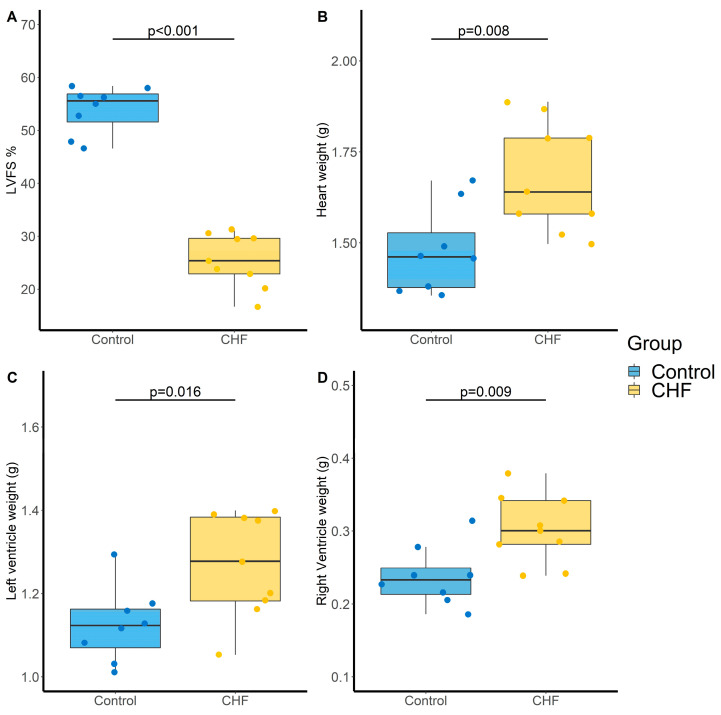
Evaluation of cardiac parameters in control and chronic heart failure (CHF) rats. (**A**) Left ventricular fractional shortening (LVFS, %), (**B**) Heart weight (g), (**C**) Left ventricle weight (g) and (**D**) Right ventricle weight (g). N = 8 and 9 for Control and CHF, respectively. *p*-values computed by Student’s *t*-test.

**Figure 4 metabolites-11-00784-f004:**
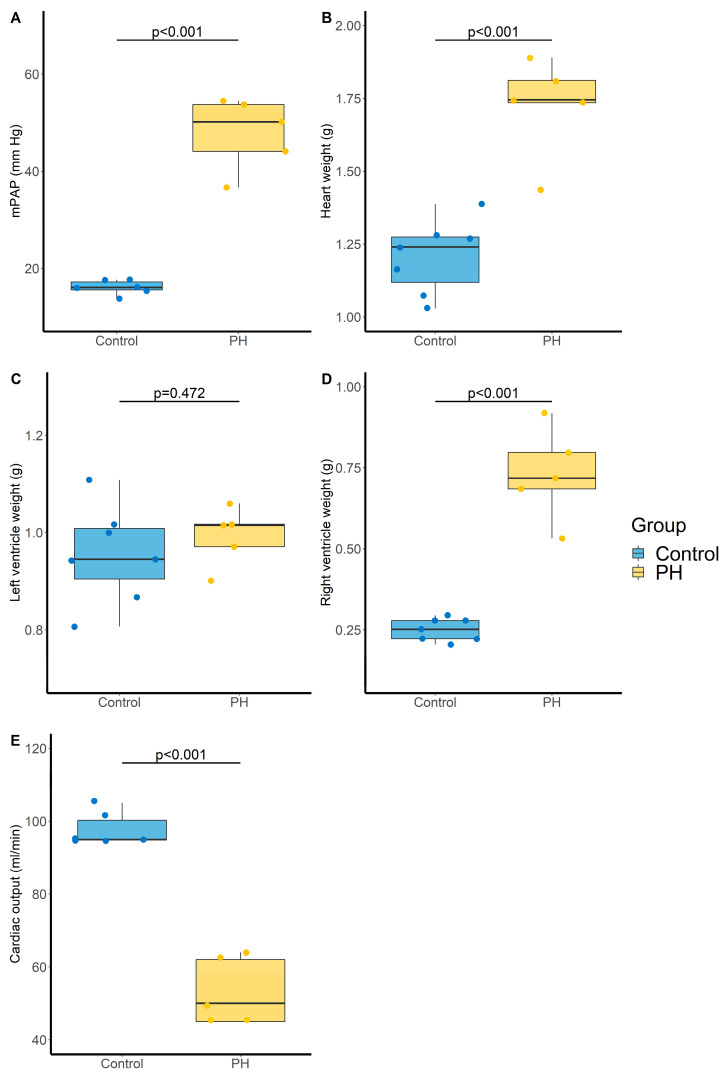
Evaluation of pulmonary and cardiac parameters in control and PH rats. (**A**) Mean pulmonary arterial pressure (mPAP, mm Hg). N = 6 and 5 for Control and PH, respectively, (**B**) Heart weight (g). N = 7 and 5 for Control and PH, respectively, (**C**) Left ventricle weight (g). N = 7 and 5 for Control and PH, respectively, (**D**) Right ventricle weight (g). N = 7 and 5 for Control and PH, respectively, and (**E**) cardiac output (mL/min). N = 6 and 5 for Control and PH, respectively. PH: pulmonary hypertension. *p*-values computed by Student’s *t*-test.

**Figure 5 metabolites-11-00784-f005:**
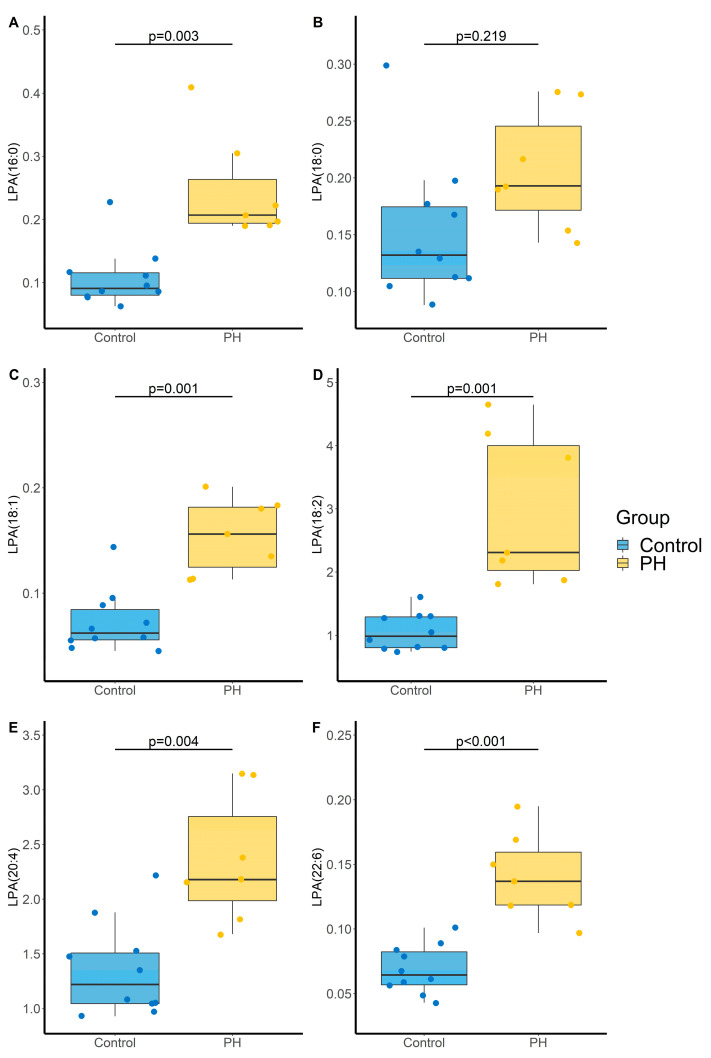
Impact of PH on lysophosphatidic acid (LPA) species (**A**) LPA(16:0), (**B**) LPA(18:0); (**C**) LPA(18:1), (**D**) LPA(18:2), (**E**) LPA(20:4), (**F**) LPA(22:6). *p*-values computed by Student’s *t*-test. N = 10 and 7 for Control and PH, respectively.

**Figure 6 metabolites-11-00784-f006:**
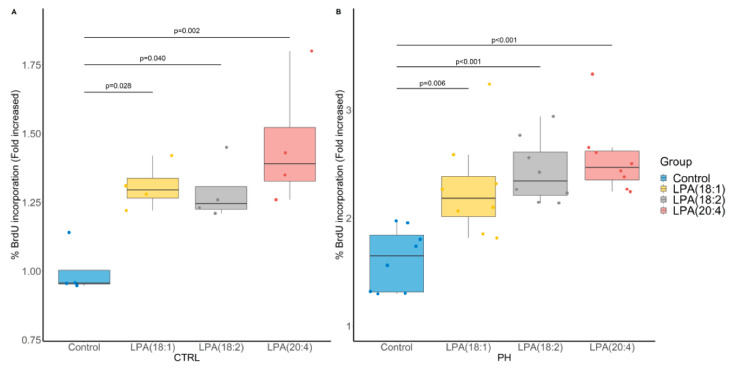
Impact of LPA species on cultured pulmonary artery smooth muscle cell (PA-SMC) proliferation assessed by 5-bromo-2-deoxyuridine (BrdU) incorporation. (**A**) PA-SMC derived from control (CTRL) patients (N = 4 in triplicates). (**B**) PA-SMC derived from a patient with pulmonary hypertension (PH) (8 replicates of 1 patient). *p*-values computed by Dunnett’s post-hoc test after ANOVA significance.

**Figure 7 metabolites-11-00784-f007:**
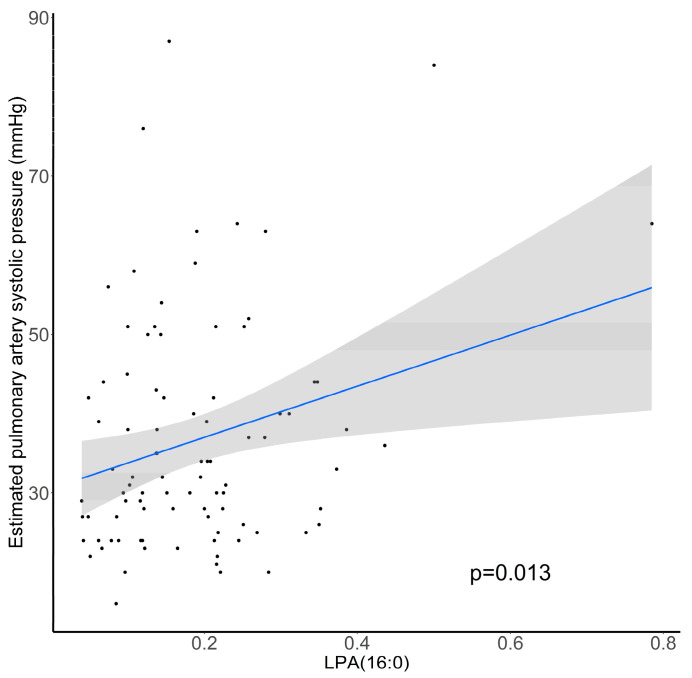
Association between estimated pulmonary artery systolic pressure (mm Hg) and relative abundance of plasma LPA(16:0). N = 90, *p*-value computed by Student’s *t*-test.

**Table 1 metabolites-11-00784-t001:** Impact of Ang-II HTN on LPA, LPL and MAG levels.

Analyte	Control Group (N = 8)	HTN Group (N = 8)	*p*-Value	Adjusted *p*-Value ^a^
LPA(16:0)	0.11 [0.11–0.12]	0.15 [0.13–0.22]	**0.0229**	0.241
LPA(18:0) ^b^	0.09 [0.08–0.10]	0.11 [0.10–0.14]	*0.0832*	0.341
LPA(18:1)	0.09 [0.09–0.10]	0.13 [0.10–0.14]	**0.0438**	0.241
LPA(18:2) ^b^	1.78 [1.57–1.97]	2.20 [1.56–2.46]	0.163	0.398
LPA(20:4) ^b^	1.91 [1.50–2.13]	2.56 [2.14–2.92]	**0.0308**	0.241
LPA(22:6) ^b^	0.10 [0.08–0.11]	0.14 [0.11–0.17]	*0.0931*	0.341
LPC(16:0)	16.7 [16.2–17.4]	17.3 [15.0–18.0]	0.568	0.781
LPC(16:1) ^b^	1.24 [1.10–1.40]	1.58 [1.40–1.77]	**0.0438**	0.241
LPC(18:0) ^b^	29.7 [27.3–32.9]	30.4 [28.7–31.7]	0.914	0.914
LPC(18:1)	7.24 [6.36–7.53]	7.22 [6.83–8.14]	0.343	0.629
LPC(18:2) ^b^	31.5 [28.5–32.6]	30.9 [28.2–32.0]	0.749	0.867
LPC(20:4) ^b^	11.1 [10.3–12.1]	12.6 [11.6–13.7]	0.148	0.398
LPC(22:6) ^b^	0.34 [0.31–0.43]	0.43 [0.38–0.46]	0.269	0.538
LPE(16:0)	0.54 [0.47–0.61]	0.62 [0.58–0.68]	0.118	0.371
LPE(18:0) ^b^	0.80 [0.76–0.93]	0.95 [0.89–1.00]	0.205	0.451
LPE(18:1) ^b^	0.22 [0.21–0.27]	0.25 [0.22–0.27]	0.376	0.636
LPE(18:2) ^b^	0.45 [0.44–0.52]	0.46 [0.41–0.49]	0.439	0.690
LPE(20:4) ^b^	0.16 [0.15–0.17]	0.17 [0.16–0.20]	0.836	0.876
LPE(22:6) ^b^	0.11 [0.09–0.12]	0.12 [0.10–0.15]	0.612	0.792
MAG(18:1)	0.13 [0.10–0.16]	0.12 [0.11–0.15]	0.805	0.876
MAG(18:2) ^b^	0.67 [0.57–1.00]	0.64 [0.58–0.79]	0.673	0.823
MAG(20:4) ^b^	0.02 [0.02–0.03]	0.02 [0.01–0.02]	0.508	0.745

^a^ *p*-values adjusted according to Benjamini and Hochberg multiple testing. ^b^ Analytes without analytical standard. HTN: hypertension, LPA: lysophosphatidic acid, LPC: lysophosphatidylcholine, LPE: lysophosphatidylethanolamine, MAG: monoacylglycerol. *p*-values computed by Student’s *t*-test. *p*-values < 0.05 are highlighted in bold.

**Table 2 metabolites-11-00784-t002:** Impact of CHF on LPA, LPL and MAG levels.

Analyte	Control Group (N = 8)	CHF Group (N = 9)	*p*-Value	Adjusted *p*-Value ^a^
LPA(16:0)	0.058 [0.046–0.060]	0.082 [0.056–0.103]	**0.025**	0.186
LPA(18:0) ^b^	0.050 [0.048–0.057]	0.070 [0.065–0.080]	0.336	0.615
LPA(18:1)	0.069 [0.063–0.076]	0.067 [0.054–0.075]	0.757	0.779
LPA(18:2) ^b^	0.87 [0.77–0.95]	1.00 [0.88–1.06]	0.193	0.499
LPA(20:4) ^b^	1.24 [1.04–1.31]	1.21 [1.14–1.56]	0.204	0.499
LPA(22:6) ^b^	0.066 [0.062–0.070]	0.068 [0.061–0.084]	0.447	0.615
LPC(16:0)	13.0 [11.3–14.2]	12.3 [11.1–13.2]	0.302	0.615
LPC(16:1) ^b^	1.06 [0.98–1.37]	0.93 [0.71–1.60]	0.750	0.779
LPC(18:0) ^b^	19.4 [15.7–20.5]	18.7 [17.1–21.4]	0.608	0.743
LPC(18:1)	5.02 [4.61–5.64]	3.97 [3.3–4.48]	**0.016**	0.186
LPC(18:2) ^b^	19.7 [18.7–21.6]	19.4 [17.1–20.5]	0.430	0.615
LPC(20:4) ^b^	8.97 [8.27–10.63]	8.4 [8.16–8.81]	0.350	0.615
LPC(22:6) ^b^	0.386 [0.331–0.393]	0.325 [0.274–0.362]	0.196	0.499
LPE(16:0)	0.55 [0.52–0.58]	0.70 [0.61–0.72]	**0.017**	0.186
LPE(18:0) ^b^	0.74 [0.63–0.79]	0.82 [0.78–0.88]	**0.046**	0.250
LPE(18:1) ^b^	0.25 [0.23–0.27]	0.27 [0.25–0.30]	0.425	0.615
LPE(18:2) ^b^	0.38 [0.37–0.42]	0.44 [0.37–0.48]	0.383	0.615
LPE(20:4) ^b^	0.158 [0.143–0.169]	0.168 [0.167–0.201]	0.089	0.390
LPE(22:6) ^b^	0.12 [0.11–0.14]	0.14 [0.12–0.15]	0.478	0.619
MAG(18:1)	0.07 [0.06–0.09]	0.11 [0.10–0.12]	0.126	0.462
MAG(18:2) ^b^	0.18 [0.15–0.21]	0.17 [0.11–0.26]	0.756	0.779
MAG(20:4) ^b^	0.0061 [0.0049–0.0084]	0.0068 [0.0045–0.0094]	0.779	0.779

^a^ *p*-values adjusted according to Benjamini and Hochberg multiple testing. ^b^ Analytes without analytical standard. HTN: hypertension, LPA: lysophosphatidic acid, LPC: lysophosphatidylcholine, LPE: lysophosphatidylethanolamine, MAG: monoacylglycerol. *p*-values computed from Student’s *t*-test. *p*-values < 0.05 are highlighted in bold.

**Table 3 metabolites-11-00784-t003:** Impact of PH induced by SuHx on LPA, LPL and MAG levels.

Analyte	Control Group (N = 10)	PH Group (N = 7)	*p*-Value	Adjusted *p*-Value ^a^
LPA(16:0)	0.091 [0.080–0.12]	0.21 [0.19–0.26]	**<0.001**	**0.003**
LPA(18:0) ^b^	0.13 [0.11–0.17]	0.19 [0.17–0.25]	0.080	0.219
LPA(18:1)	0.062 [0.056–0.085]	0.156 [0.125–0.182]	**<0.001**	**0.001**
LPA(18:2) ^b^	0.99 [0.80–1.29]	2.31 [2.03–4.00]	**<0.001**	**0.001**
LPA(20:4) ^b^	1.22 [1.05–1.51]	2.18 [1.99–2.76]	**<0.001**	**0.004**
LPA(22:6) ^b^	0.064 [0.057–0.082]	0.137 [0.119–0.160]	**<0.001**	**<0.001**
LPC(16:0)	16.8 [16.0–17.3]	16.7 [14.9–17.2]	0.357	0.604
LPC(16:1) ^b^	1.96 [1.625–2.07]	1.99 [1.585–2.86]	0.348	0.604
LPC(18:0) ^b^	32.8 [31.3–35.9]	33.7 [30.7–36.0]	0.749	0.827
LPC(18:1)	7.01 [6.66–7.32]	6.80 [6.11–7.31]	0.485	0.686
LPC(18:2) ^b^	28.7 [26.3–30.2]	28.1 [26.8–30.0]	0.669	0.827
LPC(20:4) ^b^	11.1 [10.3–11.8]	9.3 [7.6–10.4]	0.231	0.462
LPC(22:6) ^b^	0.36 [0.32–0.39]	0.36 [0.33–0.39]	0.693	0.827
LPE(16:0)	0.78 [0.73–0.79]	0.81 [0.80–1.03]	**0.015**	0.056
LPE(18:0) ^b^	1.14 [1.05–1.28]	1.39 [1.23–1.48]	0.053	0.168
LPE(18:1) ^b^	0.29 [0.27–0.34]	0.38 [0.335–0.42]	0.149	0.328
LPE(18:2) ^b^	0.53 [0.47–0.59]	0.62 [0.53–0.66]	0.457	0.686
LPE(20:4) ^b^	0.23 [0.21–0.25]	0.22 [0.18–0.24]	0.131	0.320
LPE(22:6) ^b^	0.158 [0.143–0.167]	0.176 [0.146–0.179]	0.752	0.827
MAG(18:1)	0.095 [0.066–0.156]	0.089 [0.058–0.113]	0.499	0.686
MAG(18:2) ^b^	0.70 [0.40–1.10]	0.60 [0.41–0.98]	0.860	0.860
MAG(20:4) ^b^	0.014 [0.011–0.018]	0.015 [0.006–0.023]	0.806	0.844

^a^ *p*-values adjusted according to Benjamini and Hochberg multiple testing. ^b^ Analytes without analytical standard. HTN: hypertension, LPA: lysophosphatidic acid, LPC: lysophosphatidylcholine, LPE: lysophosphatidylethanolamine, MAG: monoacylglycerol. *p*-values computed by Student’s *t*-test. *p*-values < 0.05 are highlighted in bold.

**Table 4 metabolites-11-00784-t004:** MS parameters for studied analytes.

Analyte	Mass Transition		MS Parameters			Adduct	Retention Time (min)
	m/z (MS1)	m/z (MS2)	DP (V)	CE (eV)	CXP (V)		
LPA(16:0)	409.5	152.9	−50	−40	−10	[M–H]^−^	6.8
LPA(17:1) ^a^	421.1	152.9	−50	−40	−10	[M–H]^−^	6.5
LPA(18:0) ^b^	437.3	152.9	−50	−40	−10	[M–H]^−^	7.7
LPA(18:1)	435.2	152.9	−50	−40	−10	[M–H]^−^	7.0
LPA(18:2) ^b^	433.2	152.9	−50	−40	−10	[M–H]^−^	6.3
LPA(20:4) ^b^	457.2	152.9	−50	−40	−10	[M–H]^−^	6.3
LPA(22:6) ^b^	481.2	152.9	−50	−40	−10	[M–H]^−^	6.3
LPC(16:0)	480.6	255.4	−80	−40	−10	[M–CH_3_]^−^	7.2
LPC(16:1) ^b^	478.6	253.4	−100	−40	−10	[M–CH_3_]^−^	6.3
LPC(17:1) ^a^	492.3	267.3	−80	−40	−10	[M–CH_3_]^−^	6.8
LPC(18:0) ^b^	508.4	283.5	−100	−40	−10	[M–CH_3_]^−^	8.1
LPC(18:1)	506.4	281.5	−80	−40	−10	[M–CH_3_]^−^	7.4
LPC(18:2) ^b^	504.4	279.5	−100	−40	−10	[M–CH_3_]^−^	6.7
LPC(20:4) ^b^	528.4	303.4	−100	−40	−10	[M–CH_3_]^−^	6.7
LPC(22:6) ^b^	552.4	327.4	−100	−40	−10	[M–CH_3_]^−^	6.6
LPE(16:0)	452.4	196.0	−50	−40	−10	[M–H]^−^	7.2
LPE(17:1)	464.2	196.0	−80	−40	−10	[M–H]^−^	6.9
LPE(18:0) ^b^	480.3	196.0	−50	−40	−10	[M–H]^−^	8.1
LPE(18:1) ^b^	478.3	196.0	−50	−40	−10	[M–H]^−^	7.4
LPE(18:2) ^b^	476.3	196.0	−50	−40	−10	[M–H]^−^	6.8
LPE(20:4) ^b^	500.6	196.0	−50	−40	−10	[M–H]^−^	6.7
LPE(22:6) ^b^	524.3	196.0	−50	−40	−10	[M–H]^−^	6.7
MAG(18:1)	357.2	265.0	50	17	16	[M+H]^+^	8.0
MAG(18:2) ^b^	355.2	263.0	50	17	16	[M+H]^+^	7.5
MAG(20:4) ^b^	379.3	287.1	50	17	16	[M+H]^+^	7.4

^a^ Analytes used as internal standard for area under the curve normalization. ^b^ Analytes without analytical standard. CE: collision energy, CXP: collision cell exit potential, DP: declustering potential, LPA: lysophosphatidic acid, LPC: lysophosphatidylcholine, LPE: lysophosphatidylethanolamine, MAG: monoacylglycerol.

## Data Availability

The data presented in this study are available in Appendix A.

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
