# Peer review of "Preventing the Increase in Lysophosphatidic Acids: A New Therapeutic Target in Pulmonary Hypertension?"

_metabolites, 2021, doi:10.3390/metabo11110784_

Round 1
Reviewer 1 Report
The authors have addressed all previous concerns
Reviewer 2 Report
This study, regarding circulating LPL, LPA and MAG levels in different rat models of CVD, i.e., hypertension, heart failure and pulmonary hypertension was well revised. This reviewer has no further comment.
This manuscript is a resubmission of an earlier submission. The following is a list of the peer review reports and author responses from that submission.
Round 1
Reviewer 1 Report
The work presented for review, entitled "Lysophospholipids, lysophosphatidic acids and monoacylglycerols: new therapeutic targets in cardiovascular diseases?" discusses the evolution of circulating LPL, LPA and MAG levels in three rat models of cardiovascular diseases. The topic is interesting and the choice of test compounds has been confirmed.
In accordance with the existing information on this group of diseases and the assumptions of the presented experiments, the authors wrote the manuscript correctly and carefully. The discussion is conducted in a systematic way, discussing own results against the background of the works of other authors.
The only remark concerns the used HPLC/MS2 method; has it been previously validated for the determinations of the tested substances? If so, then the missing parameters should either be supplemented or the relevant literature should be quoted.
Author Response
We would like to thank reviewer #1 for his positive comments and his relevant remark concerning the HPLC/MS2 method. In agreement, we have added a supplemental file describing the chromatographic separation of the different monitored compounds, the calibration curves of the following analytical standards: LPA(16:0), LPA(18:1), LPC(16:0), LPC(18:1), LPE(16:0) and MAG(18:1) as well as repetability and reproducibility experiments (triplicate, 2 separate days).
Please, find attached the supplemental we have added to the manuscript.
Thank you for the time you devoted to the critical and relevant review of our work.

Reviewer 2 Report
Here authors analyze the potential of Lysophospholipids, Lysophosphatidic Acids and Monoacyl-2 glycerols as a therapeutic target for CVD.
- As authors mentioned, atherosclerosis is a common denominator of CVD. Does they also measured circulating levels of lysophospholipids (LPL), lysophosphatidic acids (LPA) and monoacylglycerols (MAG) in an animal model of atherosclerosis?
- What are the levels of lipoproteins in all animal models? Is there any correlation between levels of VLDL, LDL or HDL with LPL, LPA and MAG?
- Since LPA also suggested to be protective in vascular remodeling. Does authors suggest that its a biomarker or causative factors?
Reviewer 3 Report
This study was a basic study, which aimed to examine the evolution of circulating LPL, LPA and MAG levels in different rat models of CVD, i.e., hypertension, heart failure and pulmonary hypertension in order to provide new insights to the interest of targeting LPA metabolism with pharmacological compounds. In the present study, the authors used each model of hypertension, heart failure, and pulmonary hypertension, but not other cardiovascular models. They concluded that LPA metabolism and signaling represent promising therapeutic targets that might decrease occurrence and/or improve the outcome of CVD, suggesting that modulation of LPA metabolism and signaling using specific inhibitors, antagonists or gene deletions might be of particular interest to deal with PH, where a significant increase in plasmatic LPA species was observed; the latter being associated with an increase of pulmonary artery smooth muscle proliferation. However, this reviewer considers that this paper was too preliminary and that there was nothing conclusive. This reviewer has major criticisms as described below.
Major comments:
- The title of the study. Why cardiovascular diseases? In the present study, the authors showed the significance only in pulmonary hypertension model.
- They denied the significance of circulating LPL, LPA and MAG levels in hypertension and heart failure models; however, they used only a model in each disease. To examine this kind of issue, the authors should examine patients’ blood first, and then move to animal models to examine the mechanisms.
- As described in #2, the authors should examine this issue in patients first.
- Especially in heart failure model, the authors used only myocardial infarction model, but there were a lot of causes to induce heart failure. The authors should mention the heart failure causes.
- In pulmonary hypertension model, the results were significant. However, again, how were in patients?
- Further, the authors showed only the changes of circulating LPL, LPA and MAG levels. They did not examine the function in pulmonary hypertension. For example, the authors should perform the experiments to administer these compounds or block them in animal model. The authors looked only phenomenon in animal model.
- In vitro experiment. The authors used human PA-SMCs from lung cancer, but not pulmonary hypertension. This was also a big limitation.
- In the conclusion, the authors indicated LPA metabolism and the outcome of CVD, but it was overstatement. To examine the role of LPA metabolism in CVD, the present study was too preliminary.
Reviewer 4 Report
The authors studied LPA species in three cardiac hypertrophy rat models, and showed that certain LPAs ( e.g. 18:1, 18:2, 20:4, and 22:6) are correlated with cardiac hypertrophy in PH induced by SuHx but not in the other two models in their studies. They also showed that those LPAs could induce primary smooth muscle cell proliferation in vitro. However, based on the data presented in this manuscript, it is too early to state that “LPA metabolism and signaling represent promising therapeutic targets” in their conclusion.
Comments:
- The rationale of the proposed/presented studies needs to be better provided in “Introductions”. For example, 1. why did the authors choose cardiac hypertrophy models instead of atherosclerosis mouse models while they introduce that phospholipids metabolism may play an important role in inflammation and atherosclerosis for CVD? 2. What are the pathologic differences between the 3 models that the authors have chosen in this study?
- The authors need to discuss why the significant increases in LPAs in Ang-II HTN and CHF models after BH correction. It is also important to measure/analyze key enzymes in LPA metabolisms, such as PLA/LPAAT, GPAT/LysoPL, LPP/MAGK, and ATX, in the 3 rat models in their studies to better understand the role of LPAs cardiac hypertrophy. Furthermore, instead of using “CVD”, the authors may consider addressing their results are correlated to cardiac hypertrophy?
- The animal numbers used in each group and statistical analysis methods should be shown in each figure legend. Please clarify whether all the experimental animals have been used for each figure panel. For example, in Figure 4, the number of blue dots is not consistent for all panels, and the number of yellow dots is not consistent for all panels either.
- Data in Figure 6 showed LPAs of 18:1, 18:2, and 20:4 increased primary smooth muscle cell proliferation; however, it seems like some controls still missing. Did the authors study other LPA species for primary smooth muscle cell proliferation? Did the authors study the Ang-II on smooth muscle cell proliferation? If they could not use primary cells, it is possible that using muscle cell lines?
- Lines of 119-127 are repeated.
- A typo in line 145: “… and a trend tower(d) an increase…”